# Family Physicians’ Perspectives on Their Role in Palliative Care: A Double Focus Group in Portugal

**DOI:** 10.3390/ijerph18147282

**Published:** 2021-07-07

**Authors:** Carlos Seiça Cardoso, Filipe Prazeres, Beatriz Xavier, Bárbara Gomes

**Affiliations:** 1Faculty of Medicine, University of Coimbra, 3000-370 Coimbra, Portugal; barbara.gomes@kcl.ac.uk; 2CINTESIS—Center for Health Technology and Services Research, Faculty of Medicine, University of Porto, 4200-319 Porto, Portugal; filipeprazeresmd@gmail.com; 3Faculdade de Ciências da Saúde, Universidade da Beira Interior, 6201-001 Covilhã, Portugal; 4Health Sciences Research Unit: Nursing (UICISA: E), Nursing School of Coimbra (ESEnfC), 3046-851 Coimbra, Portugal; xavier.beatriz@gmail.com; 5Cicely Saunders Institute of Palliative Care, Policy and Rehabilitation, King’s College London, London WC2R 2LS, UK

**Keywords:** primary care, palliative care, training program, focus group online

## Abstract

Background: Aggravated by the COVID-19 pandemic, the provision of palliative care for patients with palliative care needs emerges as a necessity more than ever. Most are managed in primary care by their family physicians (FP). This study aimed to understand the perspectives of specialist and trainee FPs about their role in palliative care. Methods: we conducted a double focus-group study consisting of two separate online focus-groups, one with FP specialists (*n* = 9) and one with FP trainees (*n* = 10). Results: FPs already gather two fundamental skills for the provision of palliative care: the capacity to identify patients’ needs beyond physical symptoms and the recognition that the patient belongs to a familiar, psychosocial, and even spiritual environment. They perceive their role in palliative care to be four-fold: early identification of patients with palliative care needs, initial treatment, symptom management, and patient advocacy. Participants recognized the need for palliative care training and provided suggestions for training programs. Conclusion: FPs share a holistic approach and identify multiple roles they can play in palliative care, from screening to care and advocacy. Organizational barriers must be addressed. Short training programs that combine theory, practice, and experiential learning may further the potential for FPs to contribute to palliative care.

## 1. Introduction

In the last decades, the number of people living with chronic diseases has increased, mainly due to population aging, leading to an increase in dependency status and entailing important social costs [1]. These chronic, progressive, life-threatening, and burdening diseases play an important role in this new era of the palliative care (PC) approach.

The palliative care setting is no longer restricted to oncologic terminal diseases and there are already several studies emphasizing the importance of developing palliative care in the context of chronic non-cancer diseases associated with aging, such as dementia or heart failure [2,3]. Some studies also show that there is a benefit in the early integration of palliative care, which was found in the provision of global care for patients with multiple pathologies [3,4].

Most patients with chronic illness are managed in primary care over a long period of time [5] and their family physicians (FP), can play an important role in the palliative care (PC) network; some existing evidence shows that when an FP is involved, the provision of PC seems to improve, with benefits for both patients and their families [6,7]. The prevalence of elderly patients with palliative care needs can go from 8.0–17.3%, fluctuating based on the population studied and tools used [8,9].

There are benefits of early integration of palliative care, following global patient care in several diseases [4,10], although FPs report not having the knowledge and skills to assess unmet needs [11,12].

It seems reasonable to think that strategies to help FPs get into the PC setting must be performed to help them overcome the barriers to initiate end-of-life discussions while embracing a more efficient strategy to achieve each patient’s care goals [8,12]. This study aimed to explore and summarize the perceptions of FPs and FP trainees about the role of family medicine in the Portuguese Palliative Care Network, and to summarize the best way to create and develop a training program in palliative care for FPs in Portugal.

## 2. Materials and Methods

The methodology used in the study is online focus group (OFG). The focus group methodology seems to be effective for exploring people’s opinions and experiences [13], which corresponds to the research objectives. Online focus groups are being used more often so that their participants have flexibility in answering the questions and taking part in the discussion. In addition, their use appears to be a cost- and time-saving measure, based on an accurate account of the collected data [14]. In the present study, focus groups were also conducted online to guarantee the participation of more physicians, since participation in face-to-face focus groups would have had to be limited to a time after work hours, making it difficult for physicians to attend the meetings.

### 2.1. Study Participants

To record different experiences, we conducted two separate OFGs, one for FP trainees and another for FPs. The two separate OFGs intended also to guarantee that participants would feel at ease in speaking freely and sharing their opinions. We aimed to recruit 6–10 participants per focus group to ensure discussion and manageability [13,14].

### 2.2. Inclusion Criteria

To be a Portuguese FP or FP trainee. To be integrated into the mailing list (MGF_XXI@yahoogrupos.com.br) or in an internet forum for Portuguese FPs.

### 2.3. Procedure

The invitation to participate in each OFG was shared through a mailing list and internet forums for Portuguese FPs (including both trainees and specialists). The contact information of the main author was given so that participants could report their interest to participate.

Each volunteer was asked to give their informed consent, and after, participants received information regarding the use of the OFG platform. The platform used to conduct each group was www.focusgroupit.com (accessed on 29 April 2021). Each participant had to create a profile and then they received an invitation to join each group. No compensation was offered to any of the participants.

Each participant could access other participants’ written answers in the OFG of which they were a member after they had commented on the same topic. Participants could also interact with each other’s answers.

Of the researchers, only the moderator and an assistant (with no active participation in either group) had access to the forum for collecting data. Anonymity and confidentiality of data transcription were ensured.

The study was approved by the ethics committee of the Central Regional Health Administration of Portugal (ref. nr. ARSC Estudo 19/2019).

### 2.4. Study Setting

The OFG was based on a semi-structured topic guide, developed based on a literature review and input from the researchers. A script with open-ended questions and minimal control was elaborated so that participants could discuss all relevant points of each posting.

Each OFG ran between May and June 2019, with the discussion of one major topic for 10 days, so that each participant had enough time to participate and interact with each other. Every 10 days, the moderator posted a new topic for discussion, and participants were automatically notified by email. The moderator checked the forum daily and, when necessary, asked additional questions to get a more accurate understanding of participants’ comments.

The content and timing of the posts were similar for the two OFG.

During the first one-and-a-half-week, participants were asked about the role of family physicians (FPs) in the Portuguese palliative care (PC) setting and, if they believed that FPs can participate in the PC Network, how FPs can do such, and which patients should be prioritized. The topic was presented as: “Can family physicians play a role in the provision of palliative care? If so, how can they intervene in the network? Which patients should be prioritized in family physicians’ lists?”.

In the second major topic, they were asked about PC training for FPs, if it is necessary, if it can contribute to better and earlier identification of patients with PC needs and if it would be reasonable to explore PC training in the FPs’ curriculum. The topic was presented as: “What are the main training needs in palliative care? Do you believe that training can help with earlier identification of patients with palliative care needs? Does it make sense, for example, to include training in palliative care in the curriculum of a family physician?”. Finally, participants were asked about the best and most efficient model to structure a PC training program for FPs. The topic was presented as: “What could be the most suitable way to structure, considering practical and logistical aspects, a training program for family physicians? Full days of training? Several short sessions? Theoretical lectures? Practical discussions? How to perform?”. According to the data collected from the two OFG, the analysis indicated that data saturation had been reached. Saturation was assumed once no new input about the topic was presented.

### 2.5. Analysis

The data were analyzed based on a thematic analysis created to explore FPs’ perception of their role in the PC network and how training in PC for FPs could be relevant in the Portuguese health care setting. The main themes were based on the major topics posted on the OFG platform. An inductive process was performed to code and analyze the data for the subthemes from the major themes [15]. Initial coding of the data was reviewed and discussed by the research team, which led to a more accurate coding scheme.

Separate analyses were conducted for FP trainees and FP specialists.

## 3. Results

### 3.1. Participants

We performed 2 OFG lasting 30 days each, with all the physicians that responded to the invitation (*n* = 19). One of the OFG included 10 FP trainees and the other included 9 FPs.

The first OFG included two first year, three second year, three third year, and two four year FP trainees, and 80% were female. As to the FPs’ group, four participants were specialists with up to three years of experience, and the remaining five had between three and ten years of experience; 55.6% were female.

Concerning previous experience or training in palliative care, two participants in the trainees’ OFG were master’s degree students in PC (one in the first year and one in the second), and one participant in the specialists’ OFG had a master’s in PC. Detailed information is available in Table 1.

### 3.2. The Role of FPs in the Portuguese PC Setting

The major themes were divided into sub-themes (Table 2) and are summarized below. Though the data for trainees and specialists were kept separate during the analysis, most of the summarized data is presented together and the differences between these two groups are pointed out, when relevant, across each section.

#### 3.2.1. Added Value of the Presence of Family Physicians

-Early identification of patients with palliative care needs

There was consensus among the participants from both OFG that FPs are in a privileged position to manage patients with chronic illness, and they described this role as crucial and extremely important. It was emphasized that the FP held a “privileged position in the identification and monitoring of patients with palliative care needs… demystifying concepts, intervening with families, referencing and mobilizing resources” (Participant 7, in trainees’ group).

As for the delivery of palliative care (PC) by FPs, participants shared the opinion that primary care can play a necessary and important role but that “not everyone (FPs) like[s] the subject [of palliative care] and we cannot force them” (Participant 5 in the trainees’ group). Both trainees and specialists addressed what seems to be the key point for FPs to participate in a palliative care network, mainly reinforcing their “major role in the early identification of these patients and perhaps approaching the first steps on pharmacological treatments” (Participant 6 in the trainees’ group).

-Initial treatment of patients with palliative care needs

Regarding the approach of first steps on pharmacological and non-pharmacological treatments, it is understood that the care provided to patients with chronic, progressive, and potentially fatal diseases, is different depending on the complexity of the pathology and the severity of symptoms. If, at first, the palliative care framework participated in by primary health care physicians can help patients to manage symptoms like insomnia, pain or anxiety of low-to-moderate intensity, later the progression of the disease may lead to a need for differentiation, other therapeutic approaches, and even other infrastructures.

It seems clear that FPs can play an important role in the palliative care network, and the differentiated role that these clinicians can play in the early identification of patients in need of palliative care was reinforced—“timely identification… better monitoring of patients… earlier signaling for more specific palliative care” (Participant 1 in the trainees’ group).

-Symptomatic management

Another point about which there seems to be a consensus is that FPs can play a role in symptom management for less complex patients, and it can facilitate the delivery of care to patients’ families/caregivers. As for symptomatic management, some participants suggested that the house-call consultation, as an integral part of FPs’ practice in Portugal, can be a moment of major importance to identify, treat and follow-up with patients and caregivers. “House-call consultation[s] can be very important for the end-of-life care managed in primary care” (Participant 7 in the FP’s group), being an unique and valuable moment to provide health care.

-FPs as patient advocates

The role of FPs in the PC network was understood in the context of a close articulation with different care providers “lead multidisciplinary teams, identify, support treatment and refer… accompanying the patient in a shared way with hospital colleagues” (Participant 6 in the FP’s group).

This discussion model presented some divergence with some participants in the specialists’ group arguing that some FPs may not have the interest or aptitude to be involved in a PC network—“Not everyone wants to receive training in something they do not like, they do want something practical where they can guide themselves to manage their patients” (Participant 1 in the FP’s group). They believe that the role of the FP can be assumed only in the identification and early referral of patients, leaving the management of symptoms for more differentiated care. Besides this debate, the majority of participants advocated that every doctor must have, as an “undividable part of his role as [a] doctor” (Participant 10 in the trainees’ group), the ability to manage less complex symptoms.

Training in PC that presents the main concepts of PC, the dissemination of some tools to identify and manage patients, and the basics in symptoms management will necessarily have to precede any integration of FPs in the care delivery network.

#### 3.2.2. Challenges/Barriers Perceived in Providing Palliative Care

-Complexity and diversity of patients and their conditions

It seems clear that the PC is not limited to patients with oncological diseases. All participants shared that the priorities should involve all “patients with incurable and severe diseases, in advanced or progressive stage, generating physical, psychological, social and spiritual suffering… not only patients with oncological pathology but also with neurological degenerative diseases or with advanced organ failure, including dementias and frail or dependent elderly” (Participant 9 in the trainees’ group). It was also highlighted that some “patients integrated into a unitary family or with little social support” (Participant 4 in the FP’s group) may also benefit from a PC intervention.

Another topic is related to the feasibility of this process. While it would be ideal for all patients in need of PC to be identified regarding the feasibility and transposition of such into clinical practice, some participants celebrated the idea of prioritizing the evaluation of patients through house-call consultations based on criteria that include chronic, progressive, disabling disease, or those that cause suffering or family disorder. “Patients requiring multiple medical observation[s], who suffer from chronic pathologies and resort to many consultations in a short period should also be prioritized” (Participant 6 in the trainees’ group).

Finally, some of the participants referred to the need to prioritize patients who obtain the answer “no” in the “surprise question” [16] questionnaire.

-Identifying patients

The group of trainees pointed out as the main difficulties, a lack of training as, “the preponderance of curative medicine thought”, a lack of instruments to identify patients in need of PC, and “the lack of a right time to start palliative care” (Participant 1 in the trainees’ group).

-Management of the list of patients

Specialists pointed out other barriers, besides those listed by trainees, related to the management of the list of patients. Portuguese FPs often “lose the follow-up of the most complex patients to hospital care” (Participant 7 in FP’s group), which frequently means that some of the clinical, familiar, and social context of the patient is lost in this journey. Specialists addressed what they believed to be an important issue in primary health care in Portugal—“the list of patients is oversized, and each FP has to manage and monitor many patients, which can be an obstacle” (Participant 1 in the FP’s group).

-Patients without an FP

It is also pointed out that “there are many patients without an FP, which makes it almost impossible to identify them at an earlier stage” (Participant 2 in the FP’s group) since they resort to care in the context of the exacerbation of a health problem that is often unknown at that time.

There seems to be a consensus that the main areas where work should be developed are identification, first stages’ symptomatic management, and family support, along with a process of training the general population.

### 3.3. Training in Palliative Care for FPs

#### 3.3.1. Training Needs

Belief in the need for training in PC for FPs does not seem to be universally held. While some of the participants assume, even at different levels of depth and complexity, that FPs benefit from PC training when performing their daily clinical practice (*n* = 9 in trainees’ group; *n* = 7 in FP’s group), others consider that this training should only be optional and provided for those who have some affinity for this area of knowledge.

Nevertheless, the vast majority of participants consider training necessary, and it is possible to compile the training needs identified by participants into three main cornerstones: clinical training, communication, and understanding of the network.

-Clinical training

The need to train FP on basic concepts about PC was identified. After the integration of the basics, training is needed to provide tools to identify patients with PC needs. More than definitions, it seems necessary to present validated tools that are appropriate for an FP’s clinical practice and manageable in a 15-to-20-min consultation.

Finally, participants discussed the need for training on symptom management of pain, anxiety, insomnia, constipation, dyspnea, as well as the supply of basic needs—“food, hydration and bed care” (Participant 6 in the trainees’ group). Some of the participants mentioned a greater need for training in the monitoring of the agonized patient, but this opinion was not shared by all of the participants, who believed that “this level of complexity requires the involvement of PC teams or even a PC unit” (Participant 9 in the trainee’s group).

-Communication

Communication seems to be a robust and consensual cornerstone in training needs discussed by FPs. Essentially, the need for training in “communication of bad news to the patient and family”/caregiver, training in communication skills with family/caregiver “throughout each stage of the disease, and during grieving processes” (Participant 7 in the FP’s group) were reinforced.

-Understanding of the network

Participants discussed the idea that to provide the best follow-up at any time or stage of the evolution of a patient’s disease, it is necessary to be acquainted with the available resources and infrastructures, as well as how to perform the articulation between each player involved in the network.

Respondents also reported a deficit in the baseline knowledge of how to refer a patient in need of PC who is managed in primary health care. “Which network? What is the responsiveness? Which patients should be referred and how? How to proceed with a dynamic follow-up without the patient being lost among the various doctors who take a part in the management of his disease?” (Participant 7 in the trainee’s group).

#### 3.3.2. To Build a Training Program

-Pre or postgraduate, optional or optimal

Some participants listed the training in PC as “essential both at the undergraduate level in medical schools and across all medical specialties” (Participant 9 in the trainees’ group). Although some participants argue whether this concept is a way of forcing an area of knowledge to colleagues who may not be interested, the vast majority seemed to assume that mastering the basics in PC is part of the essential skills of any clinician.

In the specific field of family medicine, this training “should not be just optional considering the high prevalence of patients with suspected need for PC managed by the FP” (Participant 9 in the trainees’ group). While some tended to reject the integration of training in PC as an integral part of the curriculum of an FP, others understood training as essential in everyday medical practice.

“FPs can contribute with a high-quality clinical practice, with the possibility to understand the patient and his family/caregiver in their natural environment and the opportunity to integrate all clinical, psychological and social data about the patient” (Participant 5 in the FP’s group), all of them being imperative characteristics to perform a better approach and management of each patient.

-Structure

A training model, in the opinion of FPs, should be short, a maximum of two days, with a theoretical framework and clarification of concepts, practical management of the patient, and communication. Thus, the structure of a training program can be summarized as follows:

Short sessions addressing theoretical concepts: Definitions and discussions of palliative care, assessment of the needs of patients, and ethical dilemmas that a clinician may confront when managing patients in need of PC. It is the opinion of the great majority of the participants that the incorporation of theoretical themes is important, but it should not occupy the majority of a training session.

Discussion of clinical cases: There was a consensus amongst participants that the best way to promote training in this area is through an interactive discussion of fictitious or real clinical cases while trying to simulate situations that could be encountered in the usual clinical practice of an FP and promoting discussion amongst the trainees. Proposed topics included: how to approach the patient, from its identification to the stratification of its clinical complexity, the management of the most common symptoms in PC, and the barriers or opportunities to its application in primary care.

Roleplays: Communication seems to be more attractive if a roleplay training-based program is the chosen methodology. By providing training on staple scenarios, simulating situations to communicate bad news to the patient or the family/caregiver in a consultation setting or in a home visit, participants can be empowered with useful communication skills.

Practical internship: It seemed to be a shared position that any training program would benefit considerably by integrating a practical internship with a team or even a Palliative Care unit into its structure. This possibility seems to make training more attractive and leaves the key message that knowledge acquisition would be more robust and effective if it would be able to put some of the learned skills into a daily clinical practice context.

## 4. Discussion

This study explores and summarizes the perceptions of Portuguese FPs about their role in the Portuguese Palliative Care Network and summarizes a way to develop a training program in palliative care for FPs.

It is known that most patients with chronic illness are managed in primary care over a long period of time, and that when the FP is involved, the provision of PC seems to improve, with benefits for both patients and their families [6,7], but it is important to understand what clinicians think and how clinicians deal with palliative care in their daily clinical practice.

Our results show that FPs seem to collect a set of fundamental skills for the provision of care at the end of life—the capacity to identify patients’ needs beyond the physical symptoms, by understanding that the patient belongs to a psychological, familiar, social, and even spiritual environment [17]; the possibility to guarantee the follow-up of the patient and their family near their environment or even near their home [17].

There was agreement amongst respondents that just as many clinicians end up collecting expertise in certain areas of family medicine, the area of palliative/end-of-life care could also stimulate the differentiation of some FPs with greater interest in the field. In fact, the Portuguese Association of Family Medicine has a study group for Palliative Care in Primary Care—GesPal.

Besides being a privileged place to perform end-of-life care, the primary health care setting may be the key to earlier identification and possibly earlier referral of patients with PC needs, which would address a major issue [18,19,20]. In the current paradigm of health care delivery in Portugal, the FP acts as an attending physician of the patient, managing both health promotion and disease prevention, which makes it difficult for any other medical specialty to meet these conditions.

Since PC training is not included in the basic curriculum of a Portuguese FP, there is a need to listen to professionals before implementing any program—specifically surveying the human resources that are available and addressing physicians interested in transporting these skills to their clinical practice while respecting those who may not feel integrated with this model of care. Nevertheless, FPs report not having the knowledge and skills to assess the unmet needs of their patients with PC needs [11,12]. There is a lack of training to identify patients in which efforts should be enhanced to prioritize curative measures and those in which comfort measures should prevail. Although there was a consensus that these comfort measures interact dynamically with curative care, it seems to be difficult to integrate these patients because the message of PC is often understood by the family and some physicians as “giving up” [21]. This identified need for training is, therefore, not only focused on health care providers but also on patients and their families to truly reach the integration of PC in the daily care paradigm, transmitting a message of a “greater investment” instead of “giving up”.

Common to both groups was the view of there being a lack of training, and both reported a need for training. Given the relatively young group of FP in our sample, we lack representation of more experienced clinicians; however, we believe that the implementation of training programs will always be more easily accepted by younger generations, which turns this research result into a valuable tool.

Some specific characteristics of primary health care that can hamper the development of a PC network involving FPs were addressed. The current high number of patients managed by each FP leads to shorter consultation times to guarantee the response and may, by itself, compromise the identification of patients in need of PC and further care delivery. The list of patients in Portugal is oversized, which makes it very difficult to carry out all tasks consistently [22], and the identification of patients in need of PC will be one of the main tasks to be lost. It was discussed that the FPs do not have much free time in their daily practice, which gets even worse due to oversized patient lists. Thus, training about how to manage a huge amount of work in a 15 to 20 min consultation turns out to be more attractive than that intended to improve communication skills. It should also be emphasized that, in addition to this barrier, the preference of many colleagues is also directed towards more clinical areas and performance algorithms and less towards empowerment in the area of communication. Unfortunately, this context of overwork by health professionals does not seem to be limited to the Portuguese context [23], and this status should be considered by those responsible for the health systems of each country.

During the discussion, a dynamic system was approached, in which patients can be managed by different health care providers and in differentiated infrastructures according to their complexity. This model approach consists of the delivery of longitudinal and bidirectional care. Thus, patients can initially be managed by FPs, and if the evolution of their disease requires referral to other health care providers or health units with greater specialization, then such can be done. However, if a patient’s symptoms/suffering alleviates, then they can again be managed by their FPs in a proximity regime and their usual social and familiar environment.

Some specificities of Portuguese FPs’ work were highlighted as important points to explore when the development of the palliative care network involves more primary care. Domiciliary consultations [24] can be the perfect moment to empower patients, families, and caregivers, dispelling myths, explaining the normal evolution of some diseases, and helping to understand the true needs and concerns brought on by the disease. The clinical evaluation can also be an essential moment to initiate and evaluate treatment results, to evaluate patients’ clinical evolution and the impact caused by the disease to patients and their families. Portugal still has a lack of training in PC and its network still needs some development [25]. This turns the empowerment of doctors, patients, and families into a difficult process, and it leads to a smaller capacity of response to some complex situations in daily activity. Nonetheless, these spotlights can be useful in helping to structure the network, meeting the professionals’ opinions and perceptions.

Another interesting point is related to the existence of patients integrated into a unitary family or with little social support. Clinicians may have to prioritize not only the identification of these patients but also manage them more closely to intervene earlier if there is an advent of chronic disease. In the past, clinicians thought of PC as a specialty associated with terminally ill patients, mostly with neoplastic pathology. Then, clinicians were able to evolve towards considering CP as measures that integrate the basic care of any patient with chronic, evolutionary, and disabling disease. In the future, clinicians can open a new window to think more carefully—and eventually try to anticipate needs—in people whose social context makes them more fragile and dependent. Clinicians already know that living alone can lead to a higher burden of disease and mortality rate [26], and this work brought the new perspective that even in the scope of palliative care, these patients may need special attention.

Either because the first steps are being taken in the construction of the PC network in Portugal, or due to the lack of information dissemination, the knowledge about the PC network seems to be an important gap in the baseline knowledge of FPs. Even in terms of the social support that a patient or family/caregiver may request during the evolution of certain disease processes, there is a lack of knowledge amongst FPs who would benefit from training. Understandably, CP training programs must be able to empower doctors about how the network works and how best to guide their patients in that same network.

At this point, it was understood that side by side with the development of some of these skills, efforts to cement the concepts and the PC network in Portugal must also be prioritized. Despite some steps already taken, there is still a long journey and many of these issues and training needs will benefit from regulation at a higher level so that they can be provided.

It was clear during the discussion that the FP is just another essential part that can contribute to the improvement of the PC network, but that even the basics on providing PC cannot be exhausted in themselves and that this PC network concept always requires a multidisciplinary approach. Most of the participants pointed to the need for structured and validated tools to identify patients with PC care needs, and this identification was one of the tasks that respondents most heavily agreed upon. There are several tools possible to be used in primary health care [27], but they still need to be made valid for the Portuguese context and its use in daily clinical practice must be tested.

Finally, participants addressed some key points that can help anyone interested in the development of a PC training program for FPs so that the professionals feel motivated to participate in the training and feel that it was designed in a way that takes into account their practical realities, expectations and identified needs. It should be considered a maximum of a two-day training program that includes short sessions addressing theoretical concepts and discussion of clinical cases, including roleplays, as well as the possibility of integrating a practical internship as a complement.

This work also has some limitations worth mentioning. Given the chosen methodology, voluntarism bias is inevitable, and it is expected that only FPs with a greater interest in the topic likely chose to participate. On the other hand, the use of an online interface can also be an important bias, since it may not reach or stimulate the participation of older and less skilled doctors given that it used computerized means. This fact is even reflected in the results, since the vast majority of FP respondents had few years of clinical practice; thus, important perspectives on the subject may have been lost. Even so, taking into account that the introduction of palliative care will take place gradually, the opinions that will have greater practical consequences will be those of professionals who will be in the medical practice in the coming years; in that case, the youth of the study’s respondents may reduce the risk of losing important information.

## 5. Conclusions

Despite some gaps in the PC network in Portugal and the lack of training for health care providers, FPs perceived themselves as competent to contribute to an earlier identification of patients with PC needs, initial management of less complex symptoms, and integration and ease of referral, depending on the complexity of the patients. FPs seem to collect a set of fundamental skills for the provision of care at the end of life and can play an important and objective role in the Portuguese palliative care network. Addressing their perspectives, major barriers on care delivery and main training needs, the network can grow through the incorporation of these professionals’ know-how. Integrating primary care in the Portuguese palliative care network may improve the delivery of care in a more effective and consistent way.

## Figures and Tables

**Table 1 ijerph-18-07282-t001:** OFG participants’ characteristics.

**FP Trainees’ OFG**			
Number	Gender	Year of residency	Training in PC
1	Female	1st year	None
2	Female	2nd year	None
3	Female	3rd year	None
4	Female	4th year	None
5	Male	1st year	None
6	Female	2nd year	1st year master’s student
7	Female	3rd years	None
8	Female	4th year	None
9	Male	2nd year	None
10	Female	3rd year	2nd year master’s student
**FP’s OFG**			
Number	Gender	Years as FP	Training in PC
1	Female	3	None
2	Female	4	None
3	Female	2	None
4	Male	2	None
5	Male	2	None
6	Male	8	None
7	Female	10	Master’s
8	Female	3	None
9	Male	3	None

**Table 2 ijerph-18-07282-t002:** Main themes and subthemes obtained through focus online groups.

**The Role of Family Physicians in Portuguese Palliative Care Setting**	
FPs in the PC network	- Early identification of patients with palliative care needs - Initial treatment of patients with palliative care needs - Symptomatic management - FPs as patient advocates
Patients and diseases	- Characteristics of patients with palliative care needs - Barriers identifying patients
**Training in Palliative Care for Family Physicians**	
Training needs	- Clinical training - Communication - Understanding of the network
To build a training program	- Concepts - Structure

## Data Availability

The study did not report any data.

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
