# Peer review of "Family Physicians’ Perspectives on Their Role in Palliative Care: A Double Focus Group in Portugal"

_ijerph, 2021, doi:10.3390/ijerph18147282_

Round 1
Reviewer 1 Report
This article is primarily expository and is clearly presented. In Line 75 the word "portuguese" must be capitalized to bring it into conformity with English grammar (thus Portuguese).
In terms of formatting I observe that starting at line 297 and ending at line 320 the spacing and font seem to be different from the rest of the text. I would recommend adjusting the fonts and spacing to make them the same throughout.
Author Response
First, I would like to thank you for your promptness in the entire editorial and review process, as well as all important comments and suggestions. I believe they were instrumental in increasing the rigor and clarity of the manuscript.
Throughout this document I will try to respond point by point to the reviewers' comments and suggestions.
“This article is primarily expository and is clearly presented. In Line 75 the word "portuguese" must be capitalized to bring it into conformity with English grammar (thus Portuguese). In terms of formatting, I observe that starting at line 297 and ending at line 320 the spacing and font seem to be different from the rest of the text. I would recommend adjusting the fonts and spacing to make them the same throughout.”
Author
Thank you very much for reading the manuscript and for the important suggestions so that it could be improved.
The manuscript underwent a process of English spell check as well as text formatting check performed by experts so that the identified errors were corrected.

Reviewer 2 Report
- Basic Reporting
The manuscript (MS) presents a clear idea of what was done and where is intended to go. Concerning the structure of the MS it seems it is well designed. Thus, the language sometimes is not quite clear. In addition, there are several typos that must be amended.
There are some line spacings that do not match each other, so some additional editing is needed.
- Experimental design
One table presenting traits from all respondents is missing.
- Validity of the findings
Considering findings there are provided adequate justifications recurring to published literature and identified some flaws related to seniority and experience considering the lack of senior FPs in the OFG sample.
The discussion is well developed.
- General comments
Lines 43-44: It should be a separate sentence.
L 47-56; 64-6; 72-3; 75-7; 87; 97-9; 124-6; 136-8; 424-6; 459-60: Unlike the above comment, here all sentences are orphans. They should not stand alone in a paragraph.
L 49; 326-7: Reference 14 should be between squared brackets as the others.
L 68-9; 108-9: ODG, FPs, PC were already explained before. So, leaving just the abbreviations seems ok.
L 108: “one-and-a-half-week” should be hyphenized as before in L 103.
L 130-1: This sentence is redundant. It was already presented in the M&M section.
L 132-5: The last sentence is redundant. In addition is more polite and better addressed to scientific purposes to use the word "gender" instead of "sex".
L 150: Authors should add a new table with basic characteristics from both OFGs groups. Namely: Participant number, participant type (trainee/student and year; FP and experience years), affinity/willingness to PC, other traits...
L 153; 170: Typo - An extra “)” should be deleted.
L 154-7: Line spacing differs form the rest of the text. Please amend accordingly.
L 169: Typo – “Trainnes” is wrongly spelled. It should be “trainees” instead. Please check this word across the main text, because is misspelled several times.
L 197: This table seems a draft. Please provide a better presentation.
L 198: This should be section 3.2.2 or so????
L 205: Typo – There is an “e” standing alone from nowhere. Please amend it accordingly.
L 242: Please substitute “his” by “their” once you do not know doctor’s gender in advance.
L 254: Attention: "Constipation" is a false friend in Portuguese to English. Please check if is "constipation" what you really mean.
L 276: Typo – Please correct to “optimal” instead.
L 322-4: It is the 3rd person singular so the sentence should be “explores and summarizes” and “summarizes”. The second “Portuguese” is redundant. “the best way” is too much presumption, it should be amended to something like “a way that suits better”.
L 328: “Clinians” is misspelled. Please amend to the right word.
L 351: The final dot “.” Is in the wrong place.
L 371: “one’s” is redundant. Please get rid of it.
L 384-7; 395: There are several “his/he” that should be changed to “their/they” accordingly.
L 401-8: Several “we” that should be changed to “clinicians” instead.
L 410: Typo - “that event” seems out of context and redundant.
L 438: Please amend “integrate” to “to integrate”.
Author Response
First, I would like to thank you for your promptness in the entire editorial and review process, as well as all important comments and suggestions. I believe they were instrumental in increasing the rigor and clarity of the manuscript.
Throughout this document I will try to respond point by point to the reviewers' comments and suggestions.
“The manuscript (MS) presents a clear idea of what was done and where is intended to go. Concerning the structure of the MS it seems it is well designed. Thus, the language sometimes is not quite clear. In addition, there are several typos that must be amended.”
“There are some line spacings that do not match each other, so some additional editing is needed.”
Author
Thank you very much for reading the manuscript and for the important suggestions so that it could be improved.
The manuscript underwent a process of English spell check as well as text formatting check performed by experts so that the identified errors were corrected.
“Experimental design
One table presenting traits from all respondents is missing.”
Author
Table 1, page 4, added.
“Validity of the findings
Considering findings there are provided adequate justifications recurring to published literature and identified some flaws related to seniority and experience considering the lack of senior FPs in the OFG sample. The discussion is well developed.”
Author
Thanks for the constructive comments.
General comments
Lines 43-44: It should be a separate sentence.
L 47-56; 64-6; 72-3; 75-7; 87; 97-9; 124-6; 136-8; 424-6; 459-60: Unlike the above comment, here all sentences are orphans. They should not stand alone in a paragraph.
L 49; 326-7: Reference 14 should be between squared brackets as the others.
L 68-9; 108-9: ODG, FPs, PC were already explained before. So, leaving just the abbreviations seems ok.
L 108: “one-and-a-half-week” should be hyphenized as before in L 103.
L 130-1: This sentence is redundant. It was already presented in the M&M section.
L 132-5: The last sentence is redundant. In addition is more polite and better addressed to scientific purposes to use the word "gender" instead of "sex".
L 150: Authors should add a new table with basic characteristics from both OFGs groups. Namely: Participant number, participant type (trainee/student and year; FP and experience years), affinity/willingness to PC, other traits...
L 153; 170: Typo - An extra “)” should be deleted.
L 154-7: Line spacing differs form the rest of the text. Please amend accordingly.
L 169: Typo – “Trainnes” is wrongly spelled. It should be “trainees” instead. Please check this word across the main text, because is misspelled several times.
L 197: This table seems a draft. Please provide a better presentation.
L 198: This should be section 3.2.2 or so????
L 205: Typo – There is an “e” standing alone from nowhere. Please amend it accordingly.
L 242: Please substitute “his” by “their” once you do not know doctor’s gender in advance.
L 254: Attention: "Constipation" is a false friend in Portuguese to English. Please check if is "constipation" what you really mean.
L 276: Typo – Please correct to “optimal” instead.
L 322-4: It is the 3rd person singular so the sentence should be “explores and summarizes” and “summarizes”. The second “Portuguese” is redundant. “the best way” is too much presumption, it should be amended to something like “a way that suits better”.
L 328: “Clinians” is misspelled. Please amend to the right word.
L 351: The final dot “.” Is in the wrong place.
L 371: “one’s” is redundant. Please get rid of it.
L 384-7; 395: There are several “his/he” that should be changed to “their/they” accordingly.
L 401-8: Several “we” that should be changed to “clinicians” instead.
L 410: Typo - “that event” seems out of context and redundant.
L 438: Please amend “integrate” to “to integrate”.
Author
The manuscript underwent a process of English spell check as well as text formatting check performed by experts so that the identified errors were corrected. Whenever the language or the phrasing construction was unclear, it was revised. Revisions to the manuscript are marked up with “Track Changes”.

Reviewer 3 Report
The article is up-to-date and very important. However, method should be characterised in details. What is more, the results should be backed up with some statistical analysis.
Author Response
First, I would like to thank you for your promptness in the entire editorial and review process, as well as all important comments and suggestions. I believe they were instrumental in increasing the rigor and clarity of the manuscript.
Throughout this document I will try to respond point by point to the reviewers' comments and suggestions.
“The article is up-to-date and very important. However, method should be characterized in details. What is more, the results should be backed up with some statistical analysis.”
Author
Thank you very much for reading the manuscript and for the important suggestions so that it could be improved.
The manuscript underwent a process of English spell check as well as text formatting check performed by experts so that the identified errors were corrected.
Methods were revised and reformulated to make them clearer. Revisions to the manuscript are marked up with “Track Changes”.
As for the suggested statistical analysis, once that the manuscript had qualitative results, the option of statistical analysis did not seem the most adequate to present the information. E

Round 2
Reviewer 2 Report
Most of the flaws in the previous version were addressed. This new version seems much better. Spellcheck was carried out as well as proofreading.
The table that was missing is now present. Its inclusion not only summarizes respondendts/interviewees, but also improves paper reading.